# FEDERATED UNLEARNING WITH DIFFUSION MODELS

## ABSTRACT

In recent years, diffusion models are widely adopted by individual users due to their outstanding performance in generation. During usage, individual users develop a need to forget privacy-related contents, making the scenario of using diffusion models on the clients a natural federated unlearning setting. For this scenario, we propose FedDUL, a **Fed**erated **UnL**earning method with **D**iffusion models, which addresses the unlearn requests from clients using the diffusion models. On one hand, we utilize local data on the clients to perform attention-based unlearning, enabling the local diffusion model to forget the concepts specified by the clients. On the other hand, we filter and group the unlearn requests from clients, gradually aggregating reasonable requests into the global diffusion model on the server, thereby protecting client privacy within the global model. The theoretical analysis further demonstrates the inherent unity between the federated unlearning problem based on diffusion models and federated learning, and extend this unity to traditional federated unlearning methods. Extensive quantitation and visualization experiments are conducted to evaluate the unlearning of both local and global models and discuss the communication and computation costs of our method, demonstrating that our method can satisfy the unlearn requests of multiple clients without compromising the generative capabilities for irrelevant concepts, providing new ideas and methods for the application of diffusion models in federated unlearning.

## 1 INTRODUCTION

In recent years, diffusion models Ho et al. (2020); Song et al. (2020); Rombach et al. (2022); Dhariwal & Nichol (2021) have gained widespread attention for their outstanding performance in generation tasks and are gradually being applied in various scenarios for individual users. The emphasis on privacy among individual users makes the use of diffusion models for generation on the clients a natural federated learning scenario. Federated learning Mammen (2021); Li et al. (2020); Acar et al. (2021); Karimireddy et al. (2020) is a decentralized machine learning approach aimed at protecting user privacy while enabling collaborative model training across devices. Many methods Yang et al. (2024); Yang et al.; Zhang et al. (2023a) have applied diffusion models to image classification or autonomous driving within federated learning.

In practical applications, the diffusion models dispatched from the server to the clients are pre-trained on vast datasets, allowing them to generate images that represent almost any distribution. This raises significant concerns regarding the potential generation of images that infringe on user privacy. This drives users to request the ability for models to unlearn these concepts, losing the capacity to generate such sensitive information to protect personal privacy and enhance user experience. This gives rise to a new federated unlearning problem setting for clients using diffusion models. Currently, some federated unlearning methods Che et al. (2023); Li et al. (2023); Cha et al. (2024) have focused on image classification tasks, but no one has yet focused on the federated unlearning problem associated with using diffusion models for image generation on the clients.

To address these challenges, this paper proposes a new problem setting, considering the forgetting needs of clients using diffusion models. Therefore, we introduce FedDUL, a **Fed**erated **UnL**earning method with **D**iffusion models. Specifically, the method includes two steps: the first step is local unlearning. When users notice that the diffusion model generates a number of images that infringe on their privacy, they can submit an unlearning request and utilize local generated data or additional provided data for local training. We design an novel attention map-based diffusion model unlearning

method. After unlearning, the local diffusion model on the client can forget privacy-related concepts while preserving the ability to generate unrelated concepts. Second, we aim to select reasonable forgetting requests from it and integrate them into the server's global model. To do this, we upload the word vectors of the sensitive concepts specified by each client along with the model parameters after forgetting training to the server. At the server, we cluster the word vectors to derive forgetting targets that represent the broader needs of users, and we aggregate the corresponding LoRA parameters into the global model, ultimately achieving a global model that satisfies the unlearning needs of various clients.

We conduct thorough theoretical analyses along with quantitative and visual experiments to demonstrate the effectiveness of our method. In the theoretical analysis, we argue that forgetting is essentially a special learning process, a transition to more general concepts. We mathematically prove this viewpoint and further extend it to traditional federated forgetting methods based on classification models. This theoretical framework provides a solid foundation for our subsequent experiments and offers a new perspective on how models learn and forget in different tasks. The experimental section evaluates the effectiveness of our proposed method from multiple aspects. First, we will conduct quantitative and visual experiments to verify our forgetting method's ability to eliminate the influence of specific concepts. We select a large number of concepts that are closely related but unrelated to the specified sensitive concepts to examine the model's generative ability regarding these unrelated concepts post-forgetting, thereby proving that our method does not impact the diffusion model's original generative capabilities. Additionally, ablation experiments will explore the impact of different methods on model performance, analyze computational complexity, and assess the contributions of various components to the final results.

In summary, the main contributions of this paper are:

- We propose a new federated unlearning problem setting for clients using diffusion models, which holds significant practical relevance as diffusion models are increasingly used by individual users.

- We introduce a novel federated unlearning method, FedDUL, for scenarios where clients utilize diffusion models. This involves a new attention map-based contrastive unlearning method for local unlearning on the clients while addressing how to aggregate multiple models that have undergone unlearning into the global model, thereby solving the unlearning requests of multiple clients simultaneously.

- We provide substantial theoretical analysis, proving the unity of the federated unlearning problem for clients using classification models or diffusion models with the federated learning problem, offering a new perspective on solving the federated unlearning problem and providing a theoretical foundation for our method.

- Comprehensive quantitative and visual experiments demonstrate that our method can satisfy the forgetting needs of multiple clients without affecting the generative capabilities regarding unrelated concepts, effectively losing the ability to generate numerous sensitive or privacy-related concepts.

## 2 RELATED WORKS

### 2.1 FEDERATED LEARNING

Federated learning was first introduced with FedAvg McMahan et al. (2017), which is characterized by its ability to aggregate knowledge from multiple clients while protecting client privacy, making model aggregation a crucial component of federated learning. McMahan et al. (2017) initially achieved model aggregation through direct parameter averaging. Subsequent methods Li et al. (2020; 2021); Karimireddy et al. (2020); Su et al. (2023); Wang et al. (2020) have followed a similar approach by aggregating model parameters without additional training. Additionally, methods like Lin et al. (2020) leverage knowledge distillation, while MOON employs client-based contrastive learning. There are also approaches based on diffusion models, such as Yang et al. (2024); Yang et al.; Zhang et al. (2023a), which contribute to the growing body of techniques in federated learning.

## 2.2 FEDERATED UNLEARNING

When federated learning methods are used in practice, it has been observed that clients may terminate collaborative learning, leading to the emergence of the federated unlearning problem Liu et al. (2022b). When clients exit, they often wish to eliminate their influence on the global model for privacy protection reasons Liu et al. (2023). Most federated unlearning methods Zhang et al. (2023b); Ding et al. (2024); Guo et al. (2023) focus on image classification scenarios, where, upon receiving an unlearning request from a client, the server needs to ensure that the global model loses the classification ability for the client-specific classes or forgets the unique sample distribution of that client. Federated unlearning can be broadly classified into two categories based on the client's involvement: active unlearning and passive unlearning Liu et al. (2023). Active unlearning Yuan et al. (2023); Liu et al. (2021); Pan et al. (2022) refers to scenarios where clients deeply engage in the unlearning process, conducting local training; some methods use knowledge distillation or pseudo-label training to achieve unlearning. In contrast, passive unlearning Zhang et al. (2023b); Jiang et al. (2024); Cao et al. (2023) involves clients not participating in the unlearning process, typically requiring the server to have retained gradients or parameters from several rounds of client uploads to facilitate unlearning. Currently, there are no federated unlearning methods specifically designed for generative models such as diffusion models, and this paper addresses this challenge.

## 2.3 DIFFUSION MODEL

Diffusion models are first introduced in Sohl-Dickstein et al. (2015), and subsequent advancements such as the latent diffusion model proposed by Ho et al. (2020), along with sampling methods like DDIM Song et al. (2020), PNDM Liu et al. (2022a), have achieved impressive results, establishing diffusion models as a mainstream choice for generative models. Stable Diffusion Rombach et al. (2022) has further ignited a trend in AIGC (Artificial Intelligence Generated Content). A significant highlight of diffusion models is their conditional generation capability, which allows them to generate data from desired distributions given appropriate conditions, such as images Saharia et al. (2022a); Wang et al. (2022); Su et al. (2022b); Zhang & Agrawala (2023), text Nichol et al. (2021); Saharia et al. (2022b); Kim et al. (2022); Preechakul et al. (2022), or models Dhariwal & Nichol (2021); Feng et al. (2022); Xie et al. (2023). However, since diffusion models are typically pre-trained on vast datasets covering a wide range of distributions, they can also generate images containing sensitive content, including privacy-related information or graphic and explicit material Kumari et al. (2023). As a result, the concept erasure in diffusion models has garnered increasing attention. Research on concept erasure is still in its early stages. Some existing methods utilize foundation model-assisted LoRA fine-tuning Lu et al. (2024), contrastive learning Gandikota et al. (2024); Li et al. (2024), or fine-tuning of text encoders Zhang et al. (2024) to achieve the forgetting of specified concepts. While some of these methods share similarities with the local unlearning approach used in this paper—particularly in employing attention mechanisms for unlearning training—they do not address the federated unlearning problem setting. These methods typically focus on forgetting one concept at a time. For example, Lu et al. (2024) implements the forgetting of multiple concepts by using separate model parameters for each concept. In contrast, the federated unlearning problem setting with diffusion models proposed in this paper employs a single global model to simultaneously forget multiple concepts while protecting client privacy.

## 3 METHOD

In this section, we first introduce our problem setting in the Preliminaries, where we also define various symbols. Next, we detail our method, which consists of two main parts: first, the local unlearning process conducted by the clients, and second, the model aggregation on the server. Finally, in the theoretical analysis section, we demonstrate the unity of the federated unlearning problem based on diffusion models with the federated learning problem, further extending this conclusion to traditional federated forgetting methods based on classification models. The overall framework of our method is illustrated in Figure 1, and the pseudo code of our method is provided in the appendix.

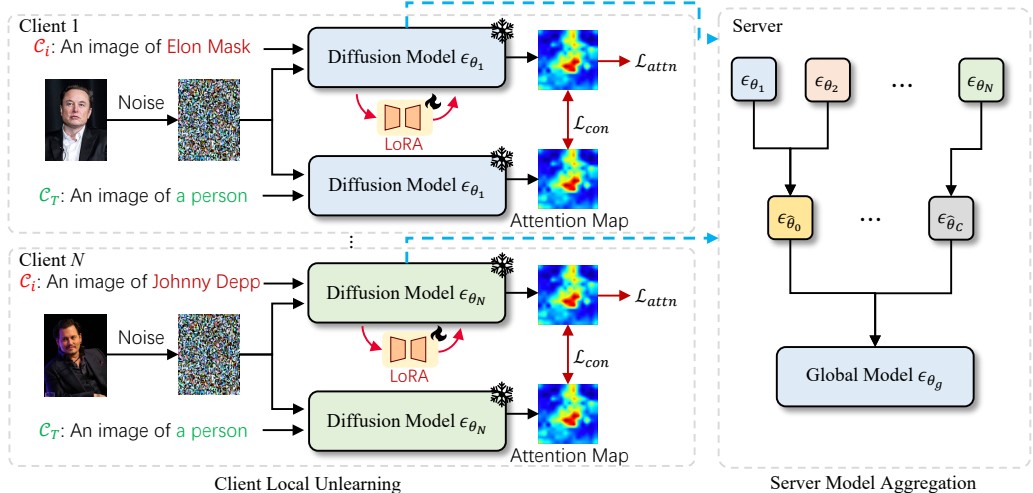

Figure 1: The overall framework of FedDUL, including two main parts: Client Local Unlearning and Server Model Aggregation. Firstly, each client uses local data along with the defined unlearning concept $\mathcal{C}_i$ and the corresponding target concept $\mathcal{C}_T$ to perform LoRA fine-tuning on the local diffusion model $\epsilon_{\theta_i}$, thereby achieving unlearning of privacy-related concepts. The trained parameters are then uploaded to the server. The server clusters the received concept word embeddings and performs two rounds of parameter averaging to obtain the global model based on the clustering results.

## 3.1 PRELIMINARIES

### 3.1.1 DIFFUSION MODELS.

The DMs study the transformation from the Gaussian distribution to the realistic distribution by iterative denoising. During sampling, the DM $\epsilon_\theta$ samples $\mathbf{s}_T$ from the Gaussian distribution, where $T$ is the predetermined maximum timestep. The DM takes $\mathbf{s}_T$ as the initial noise of the denoising process and uses the input text prompt $p$ and the input image $q$ as conditions. After $T$ timesteps of denoising, $\mathbf{s}_T$ is restored to a real image $\mathbf{s}_0$ with specified semantics. For any given time step $t \in \{0, \ldots, T\}$, the sampling process is as follows:

$$\mathbf{s}_{t-1} = \sqrt{\alpha_{t-1}}\left(\frac{\mathbf{s}_t - \sqrt{1-\alpha_t}\epsilon_\theta(\mathbf{s}_t, t|p, q)}{\sqrt{\alpha_t}}\right) + \sqrt{1 - \alpha_{t-1} - \sigma_t^2} \cdot \epsilon_\theta(\mathbf{s}_t, t|p, q) + \sigma_t \varepsilon_t \quad (1)$$

where $\alpha_t$, $\alpha_{t-1}$ and $\sigma_t$ are pre-defined parameters, $\varepsilon_t$ is the Gaussian noise randomly sampled at each timestep.

### 3.1.2 PROBLEM SETTING AND NOTATION

This paper considers a new federated unlearning problem setting when the client models are diffusion models. We assume there are $N$ clients, and the server first distributes the diffusion model $\epsilon_{\theta_p}$ pre-trained on a vast dataset $\mathcal{D}_p$ to these clients. Each client $i$, during usage, develops a unlearning request regarding a unlearning concept $\mathcal{C}_i$ and thus provides a dataset $\mathcal{D}_i$ related to the unlearning concept $\mathcal{C}_i$ for local unlearning. Ultimately, these local unlearning processes are aggregated into a global model $\epsilon_{\theta_g}$, following the global objective function:

$$\max_{\theta_g} \frac{1}{N} \sum_{i=0}^{N} D_{KL}(p_{\theta_g}(s_t|\mathcal{C}_i)||p_{\theta_p}(s_t|\mathcal{C}_i)), \forall t \in \{0, ..., T\} \quad (2)$$

where $D_{KL}$ represents the KL divergence, the initial noise of denoising process $s_T \sim \mathcal{N}(0, \mathcal{I})$, $p_{\theta_g}(s_t|\mathcal{C}_i)$ represents the conditional distribution of the global model $\epsilon_{\theta_g}$ conditioned on the unlearning concept $\mathcal{C}_i$, and $p_{\theta_p}(s_t|\mathcal{C}_i)$ represents the similar conditional distribution of the pre-trained model

$\epsilon_{\theta_p}$. From the loss function, it is clear that the goal of our problem setting is to obtain a global model $\epsilon_{\theta_g}$ that maximizes the KL divergence between the conditional distributions regarding the unlearning concept $\mathcal{C}_i$ and satisfies each client's unlearning request, meaning it does not related to each client's privacy and does not include the sensitive information requested by the clients. We will also conduct quantitative performance testing of the model according to this objective in the subsequent experimental section.

### 3.2 LOCAL UNLEARNING

As mentioned before, after the server distributes the pre-trained model $\epsilon_{\theta_p}$ to the clients, the clients may find that the diffusion model, having been pre-trained on a vast pre-trained dataset $\mathcal{D}_p$, could generate outputs that infringe on client privacy or contain sensitive information the clients wish to avoid. In this case, clients need to perform unlearning training using local data $\mathcal{D}_i$ to ensure the diffusion model loses the ability to generate such privacy-related or sensitive information.

Specifically, the goal of unlearning does not mean generating noises or meaningless outputs when given the unlearning concept $\mathcal{C}_i$. Instead, it involves finding a target concept $\mathcal{C}_T$ that are more generalized and do not infringe on user privacy or include sensitive information. For instance, if a client deems the generation of a specific person as a privacy violation, the unlearning objective would be to forget the specific appearance of that individual and instead generate a more generalized people. Thus, for each client's unlearning concept $\mathcal{C}_i$, users can define a more generalized target concept $\mathcal{C}_T$, and we define the client's unlearning objective as training a local model $\epsilon_{\theta_i}$ to minimize the KL divergence between the two conditional distributions:

$$\min_{\theta_i} D_{KL}(p_{\theta_i}(s_t|\mathcal{C}_i)||p_{\theta_p}(s_t|\mathcal{C}_T)), \forall t \in \{0, ..., T\} \tag{3}$$

We achieve this objective using two attention map-related loss functions. For the same initial noise $s_T \sim \mathcal{N}(0, \mathcal{I})$, we guide the denoising process using both the unlearning concept $\mathcal{C}_i$ and the target concept $\mathcal{C}_T$. At each time step of the denoising process, we obtain two attention maps corresponding to the current denoised sample $s_t$. To ensure these two conditional distributions are as close as possible, we take the contrastive loss between the two attention maps as our loss function:

$$\mathcal{L}_{con} = \sum_l \left\| A_t^l(\mathcal{C}_i) - A_t^l(\mathcal{C}_T) \right\|^2, \forall t \in \{0, ..., T\} \tag{4}$$

where $l$ is the layers of the local model $\epsilon_{\theta_i}$, $A_t^l(\mathcal{C}_i)$ and $A_t^l(\mathcal{C}_T)$ are the attention maps conditioned on the unlearning concept $\mathcal{C}_i$ and the target concept $\mathcal{C}_T$ of layer $l$ at timestep $t$. It's important to note that we do not use the distance between the results of the two denoising processes as the loss function; instead, we focus on the attention maps. This is because the noise content in the denoising results at most time steps is quite high. Although the starting point of the denoising process is equivalent, the randomness of the denoising limits how much the initial noise can be denoised at the given timestep, leading to limited semantic information from the distances between the outputs and potentially introducing error information. In contrast, attention maps represent the contours and key points of the corresponding semantics found in the current image, significantly reducing noise content. Finding closeness between attention maps can also yield better learning outcomes, which we demonstrate in subsequent ablation experiments.

However, simply bringing the attention maps of the two concepts closer does not ensure the effectiveness of unlearning. Generally, the target concept $\mathcal{C}_T$ is a more generalized version of the unlearning concept $\mathcal{C}_i$, and their semantics often exhibit a hierarchical relationship with strong correlations. While making the two attention maps closer, we also need to minimize the guiding influence of the unlearning concept $\mathcal{C}_i$ on generation. Therefore, we introduce another attention map-related loss function, using the mean attention value of the sensitive concept at the current time step as the loss function, aiming for minimal activation values caused by the sensitive concept:

$$\mathcal{L}_{attn} = \sum_l \left\| A_t^l(\mathcal{C}_i) \right\|_F^2, \forall t \in \{0, ..., T\} \tag{5}$$

The overall loss function is as follows:

$$\mathcal{L} = \mathcal{L}_{con} + \beta\mathcal{L}_{attn} \tag{6}$$

where $\beta$ is the weight of $\mathcal{L}_{attn}$. Through this loss, we complete the unlearning process on the client's local diffusion model, enabling the generation of images representing a more generalized target concept upon receiving a sensitive concept. However, in a federated unlearning problem setting, it is crucial to address how to aggregate these locally trained results into the global model, without revealing the exact unlearning concepts or providing any data to the server. We primarily tackle this issue in the subsequent model aggregation section.

### 3.3 MODEL AGGREGATION

To integrate the unlearning requests of various clients into the global model $\epsilon_{\theta_g}$, we need an additional model aggregation step. First, the server will receive the trained model parameters $\epsilon_{\theta_i}, i \in \{0, ..., N\}$ uploaded by the clients along with the word embeddings for the unlearning concepts $\mathcal{C}_i, i \in \{0, ..., N\}$. To avoid performance degradation from aggregating a large number of models simultaneously, we cluster these word embeddings based on their similarities. Each cluster contains word embeddings that share similar information. We use a hierarchical structure to first aggregate the model parameters corresponding to the word embeddings within each cluster, resulting in the model parameters for the overall cluster, and then further aggregate the cluster parameters to obtain the final global model. We use the averaging method for model parameter aggregation:

$$\epsilon_{\theta_g} = \sum_{c=0}^{C} \frac{1}{C}\epsilon_{\hat{\theta}_c}, \epsilon_{\hat{\theta}_c} = \sum_{\epsilon_{\theta_i} \in M_c} \frac{1}{|M_c|}\epsilon_{\theta_i}, c \in \{0, ..., C\} \tag{7}$$

where $C$ is the number of clusters and $M_c$ is a cluster of the client local model $\epsilon_{\theta_i}$. It's important to note that, intuitively, in the federated learning process, each model contains unique knowledge that is not influenced by the others, and averaging allows for the aggregation of this unique knowledge into the global model. However, in the unlearning process, each model has forgotten a portion of its knowledge. During aggregation, it may seem that the knowledge could complement one another, resulting in the global model not actually achieving unlearning. While this reasoning is fundamentally flawed, as unlearning and learning processes are essentially unified. We demonstrate this point in subsequent theoretical analysis and experimental sections.

### 3.4 THEORETICAL ANALYSIS

In this section, we explore the unity between our proposed federated unlearning problem setting for clients using diffusion models and the federated learning problem setting, and extend this conclusion to traditional federated unlearning problems for classification models.

First, we consider the federated unlearning problem setting based on diffusion models. As analyzed previously, we can transform the maximization objective function from Eq. (2) into a minimization objective function similar to Eq. (3):

$$\min_{\theta_g} \frac{1}{N} \sum_{i=0}^{N} D_{KL}(p_{\theta_g}(s_t|\mathcal{C}_i)||p_{\theta_p}(s_t|\mathcal{C}_T)), \forall t \in \{0, ..., T\} \tag{8}$$

Thus, we arrive at the following theorem:

**Theorem 1** *For the federated unlearning problem with the objective function as in Eq. (8), the problem setting exhibits unity with federated learning, meaning its loss function can be expressed as:*

$$\min_{\theta_g} \sum_{i=0}^{N} \mathcal{L}_i(\theta_g), where \mathcal{L}_i(\theta_g) = \sum_{s_T \sim \mathcal{N}(0,I), t \in \{0,..,T\}} \log p_{\theta_p}(s_t|\mathcal{C}_T)(p_{\theta_i}(s_t|\mathcal{C}_i) - p_{\theta_g}(s_t|\mathcal{C}_i))$$
$$+ H(p_{\theta_p}(s_t|\mathcal{C}_i)) - H(p_{\theta_i}(s_t|\mathcal{C}_i)) \tag{9}$$

A detailed proof of the theorem can be found in the appendix. This demonstrates the essential unity between the federated unlearning problem based on diffusion models and the federated learning problem.

This conclusion can also be extended to traditional federated unlearning problems for classification models. Similar to the definition in Eq. (2), for the global model before unlearning $\theta_p$ in the traditional federated unlearning problem, the objective function can be defined as:

$$\max_{\theta_g} \frac{1}{N} \sum_{i=0}^{N} \sum_{x_j \in \mathcal{D}_i} D_{KL}(p_{\theta_p}(x_j) || p_{\theta_g}(x_j)) \tag{10}$$

where $\mathcal{D}_i$ is the unlearning set of client $i$. Based on this objective function, firstly, we can obtain the following lemma:

**Lemma 1** *For each client $i$ and its unlearning set $\mathcal{D}_i$, the objective function in Eq. (10) has an upper bound, and the upper bound is achieved when $p_{\theta_g}(x_j)$ and $x_j$ are independent for all $x_j \in \mathcal{D}_i$, for example:*

$$D_{KL}(p_{\theta_p}(x_j) || p_{\theta_g}(x_j)) \leq D_{KL}(p_{\theta_p}(x_j) || \mathcal{N}(0, I)), \forall x_j \in \mathcal{D}_i \tag{11}$$

Based on Lemma 1, similarly, we can obtain the following theorem:

**Theorem 2** *For the traditional federated unlearning problem with the objective function as in Eq. (10), the problem setting exhibits unity with federated learning, meaning its loss function can be expressed as:*

$$\min_{\theta_g} \sum_{i=0}^{N} \mathcal{L}_i(\theta_g), where \mathcal{L}_i(\theta_g) = \sum_{x_j \in \mathcal{D}_i, \varepsilon \sim \mathcal{N}(0, I)} p_{\theta_p}(x)(\log p_{\theta_i}(x) - \log \varepsilon) \tag{12}$$

A detailed proof of the theorem can be found in the appendix. Theorems 1 and 2 demonstrate the unity between federated unlearning and federated learning problems, indicating that we can leverage traditional federated learning methods to address federated unlearning issues. This provides a theoretical foundation for using model aggregation techniques in federated unlearning problem, such as knowledge distillation, client training with pseudo-label, and the averaging methods discussed in this paper. Additionally, it offers a fresh perspective for tackling federated unlearning challenges.

## 4 EXPERIMENTS

### 4.1 EXPERIMENTAL SETTINGS

#### 4.1.1 DATASET

We mainly simulate two potential privacy infringement scenarios for clients: one involving the generation of a specified face, and the other involving the generation of a specified artist's style.

**Celebs** For the dataset of faces, we collect faces of 100 celebrities along with 20 facial photos of each celebrity from the internet. We select 50 celebrities as the Unlearn Set, representing 50 clients, while the remaining 50 celebrities formed the Retain Set, which was used to test the generation capability for unrelated categories.

**Artists** For the artist style dataset, we gather 300 kinds of artist styles that Stable Diffusion can generate from the Image Synthesis Style Studies Database Ima, and collected 20 representative works for each artist from WikiArt Mancini et al. (2018). We also select 150 celebrities as the Unlearn Set, representing 150 clients, while the remaining 150 celebrities formed the Retain Set.

Due to space constraints, detailed descriptions of the datasets and additional implementation details can be found in the supplementary material and appendix.

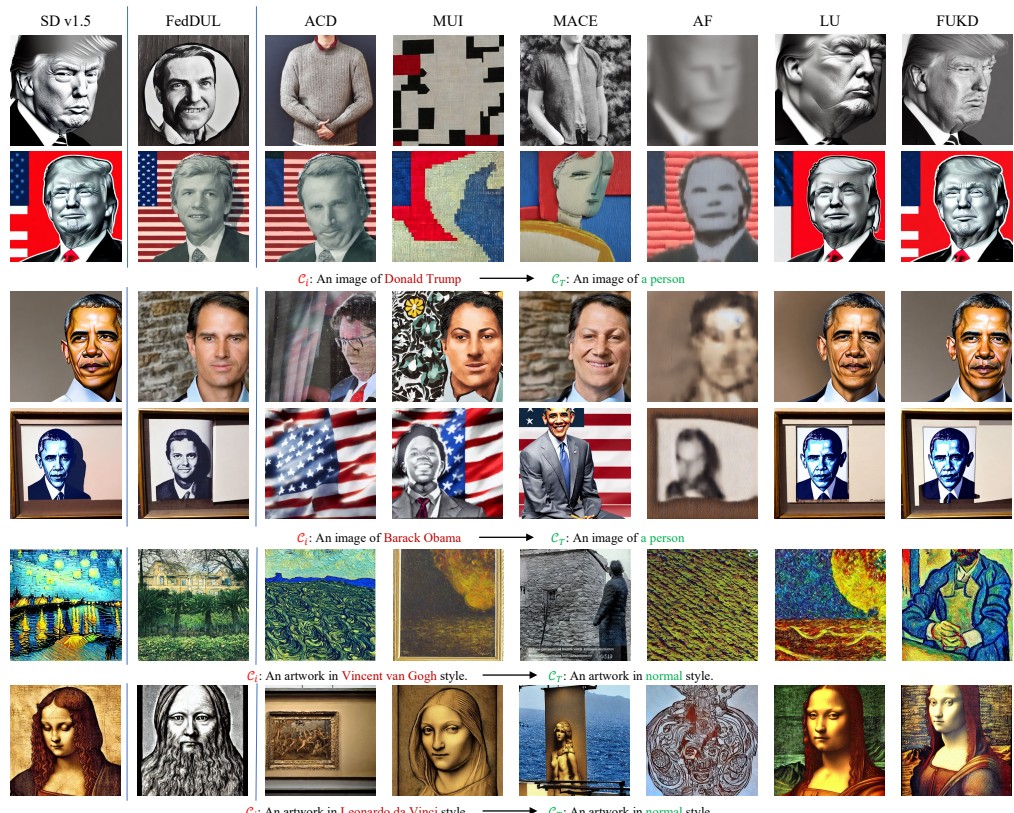

Figure 2: The visualization results of unlearning concepts for different methods.

### 4.1.2 EVALUATION METRICS

We primarily select two quantitative metrics to evaluate the quality of unlearning. First, we directly compare the KL divergence between the generated dataset and the original dataset. For unlearning concepts, a larger KL divergence indicates better unlearning performance, whereas for unrelated concepts, a smaller KL divergence is preferable. Second, we train classification models separately on the generated and original datasets and tested their accuracy on a generated test dataset. For unlearning concepts, lower classifier accuracy indicates better unlearning, while for unrelated concepts, higher classifier accuracy is preferred.

### 4.1.3 COMPARED METHODS

When comparing the effects of local unlearning at the clients, we select three concept erasing methods for diffusion models as baselines, **ACD** Kumari et al. (2023), **MUI** Li et al. (2024) and **MACE** Lu et al. (2024). For the comparison of global models, we provide results for the local model, the global model obtained by direct averaging, and the global model obtained by our proposed cluster-based averaging. For example, in ACD, **ACD-L** represents the results generated by the local model trained using ACD, **ACD-A** represents the results from directly averaging the local models, and **ACD-C** represents the results from averaging after clustering.

Since most existing federated unlearning methods are not applicable to our newly proposed problem setting, we adapt three methods for diffusion models and conducted experiments: **AF** Li et al. (2023), **LU** Cha et al. (2024), and **FUKD** Wu et al. (2022). Specifically, for **AF** and **LU**, which are based on pseudo-label training, we provide the expected labels for the clients' images and used these image-label pairs to train the clients' diffusion models. For FUKD, after the client completed unlearning

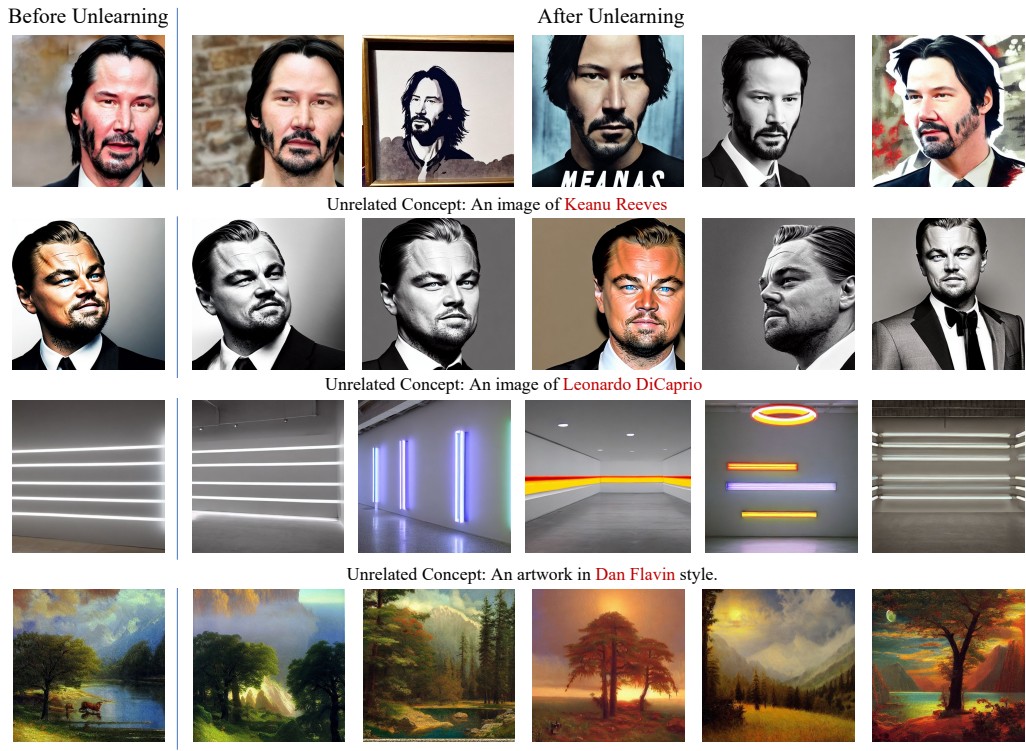

Before Unlearning | After Unlearning

Unrelated Concept: An image of Keanu Reeves

Unrelated Concept: An image of Leonardo DiCaprio

Unrelated Concept: An artwork in Dan Flavin style.

Unrelated Concept: An artwork in Albert Bierstadt style.

Figure 3: The visualization results of retain concepts for different methods.

| | | | ACD-A | ACD-C | MUI-A | MUI-C | MACE-A | MACE-C | AF | LU | FUKD | FedDUL |
|---|---|---|---|---|---|---|---|---|---|---|---|---|
| Celebs | Unlearn Set | KLD↑ | 138.12 | 164.91 | 142.88 | 175.19 | 145.14 | 176.83 | 127.3 | 115.35 | 146.65 | **187.27** |
| | | Acc.↓ | 65.87 | 43.86 | 62.21 | 39.5 | 59.44 | 35.58 | 72.04 | 75.25 | 67.61 | **34.88** |
| | Retain Set | KLD↓ | 71.93 | 75.56 | 73.35 | 77.01 | 74.92 | 73.12 | 106.29 | 125.73 | 98.71 | **68.65** |
| | | Acc.↑ | 74.6 | 76.03 | 76.84 | 72.52 | 73.32 | 75.06 | 62.09 | 58.66 | 67.62 | **79.06** |
| Artists | Unlearn Set | KLD↑ | 174.08 | 193.32 | 168.53 | 182.34 | 171.85 | 183.87 | 154.21 | 148.91 | 176.8 | **205.44** |
| | | Acc.↓ | 37.16 | 33.25 | 41.82 | 35.74 | 37.62 | 34.48 | 45.3 | 47.27 | 36.83 | **31.8** |
| | Retain Set | KLD↓ | 103.23 | 106.18 | 98.81 | 101.36 | 96.71 | 94.52 | 143.09 | 124.01 | 105.68 | **89.79** |
| | | Acc.↑ | 47.48 | 43.83 | 52.03 | 50.12 | 53.32 | 55.43 | 34.48 | 37.92 | 41.55 | **62.25** |

Table 1: The performance of different methods on the Celebs and Artists datasets for the Unlearn Set and Retain Set. KLD represents the KL divergence between the generated dataset and the original local dataset. Acc. denotes the accuracy of the classifier trained on the generated dataset. ↑ indicates that a higher value is better for the given metric, while ↓ indicates that a lower value is better. The best performance for each metric is highlighted in bold.

training, the client's images were uploaded to the server for knowledge distillation on the global model.

## 4.2 MAIN RESULTS

Table 1 shows the performance of our method and various compared methods on four datasets. We highlight several observations:

- For both the Celebs dataset and the Artists dataset, our method achieved the best performance on both the KLD and accuracy (Acc.) metrics for the Unlearn Set and Retain Set. This indicates that our method effectively maintains the generation capability for unrelated categories while obtaining a global model that satisfies all clients' unlearning requests.

| $\mathcal{L}_{con}$ | $\mathcal{L}_{attn}$ | Cluster-based Aggregation | Celebs | | | | Artists | | | |
|---|---|---|---|---|---|---|---|---|---|---|
| | | | Unlearn Set | | Retain Set | | Unlearn Set | | Retain Set | |
| | | | KLD↑ | Acc.↓ | KLD↓ | Acc.↑ | KLD↑ | Acc.↓ | KLD↓ | Acc.↑ |
| ✓ | | | 165.49 | 43.47 | 82.24 | 72.29 | 186.56 | 37.69 | 102.61 | 56.38 |
| | ✓ | | 162.61 | 39.98 | 73.42 | 75.64 | 188.08 | 36.78 | 96.92 | 58.44 |
| ✓ | ✓ | | 174.18 | 36.96 | 71.41 | 77.87 | 192.31 | 34.66 | 93.06 | 60.82 |
| ✓ | ✓ | ✓ | **187.27** | **34.88** | **68.65** | **79.06** | **205.44** | **31.8** | **89.79** | **62.25** |

Table 2: The influence of different conditions in FedDUL.

- For the ACD, MUI, and MACE methods, the global models obtained using the clustering-based average aggregation method outperformed those obtained through direct parameter averaging. This demonstrates that direct averaging can diminish the unlearning effects at the global level, especially with a large number of clients, highlighting the effectiveness of our proposed model aggregation approach.

- FUKD showed relatively good performance, but since it directly used clients' data for knowledge distillation, it compromises user privacy, making it less practical for real-world use.

- The AF and LU methods exhibited poor performance on the Retain Set, mainly because they used incorrect pseudo-labels for training the diffusion models, which failed to protect the generation capability for unrelated categories.

We also conduct extensive visualization experiments, as shown in Figure 2 and 3. From these results, it can be observed that our method successfully obtains a global model that meets the unlearning requests of each client, while maintaining the generation capability for unrelated concepts with minimal impact.

### 4.3 ABLATION EXPERIMENTS

We conduct ablation experiments on several components of our method, including the two loss functions and the cluster-based model aggregation. The results are shown in Table 2. From the table, it can be seen that removing $\mathcal{L}_{con}$, $\mathcal{L}_{attn}$, or Cluster-based Aggregation has a significant influence on the performance.

### 4.4 LIMITATIONS AND DISCUSSIONS

In this section, we discuss factors that limit the practicality of our method, including communication cost and computation cost. Regarding communication cost, since we use LoRA fine-tuning during the local unlearning process on clients, only the parameters from the LoRA layers need to be uploaded, resulting in a very limited communication cost. Moreover, as the unlearning process does not require multiple rounds of communication, the communication cost of our method is similar to most one-shot federated learning methods using LoRA fine-tuning, which means that communication cost does not significantly limit the practicality of our method. Regarding computation cost, we must acknowledge that since LoRA fine-tuning of the diffusion model is needed on clients, our method requires enough computational capability on the clients. However, given that some existing methods already involve training diffusion models Yang et al. or other foundational models Su et al. (2022a); Guo et al. (2022) on clients, the computational cost is also a limited constraint on the practicality of our method.

## 5 CONCLUSION

In this paper, we extend the federated unlearning problem to the scenario where clients use diffusion models and propose FedDUL, introducing a novel client local unlearning method and exploring how to aggregate a large number of unlearned models into a global model. To demonstrate the effectiveness of our method, we construct two datasets, and conduct extensive quantitative and visual experiments. Additionally, through theoretical analysis, we prove the inherent unity between federated unlearning and federated learning, offering a new perspective on solving the federated unlearning problem and providing a theoretical foundation for our method.

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

# A APPENDIX

## A.1 PROOFS

**Theorem 1** *For the federated unlearning problem with the objective function as in Eq. (8), the problem setting exhibits unity with federated learning, meaning its loss function can be expressed as:*

$$\min_{\theta_g} \sum_{i=0}^{N} \mathcal{L}_i(\theta_g), where \mathcal{L}_i(\theta_g) = \sum_{s_T \sim \mathcal{N}(0,I), t \in \{0,..,T\}} \log p_{\theta_p}(s_t|\mathcal{C}_T)(p_{\theta_i}(s_t|\mathcal{C}_i) - p_{\theta_g}(s_t|\mathcal{C}_i))$$
$$+ H(p_{\theta_p}(s_t|\mathcal{C}_i)) - H(p_{\theta_i}(s_t|\mathcal{C}_i)) \tag{13}$$

*Proof.* For each client, the objective function of local unlearning is :

$$\min_{\theta_i} D_{KL}(p_{\theta_i}(s_t|\mathcal{C}_i)||p_{\theta_p}(s_t|\mathcal{C}_T)), \forall t \in \{0, ..., T\} \tag{14}$$

Considering all the clients, the global objective function is:

$$\min_{\theta_0,...,\theta_N} \frac{1}{N} \sum_{i=0}^{N} D_{KL}(p_{\theta_i}(s_t|\mathcal{C}_i)||p_{\theta_p}(s_t|\mathcal{C}_T)), \forall t \in \{0, ..., T\} \tag{15}$$

Meanwhile, as defined in Eq. (8) the objective function of global model $\epsilon_{\theta_g}$ is:

$$\min_{\theta_g} \frac{1}{N} \sum_{i=0}^{N} D_{KL}(p_{\theta_g}(s_t|\mathcal{C}_i)||p_{\theta_p}(s_t|\mathcal{C}_T)), \forall t \in \{0, ..., T\} \tag{16}$$

Formally, we can prove that the objective function in Eq. (16) has a lower bound, and this lower bound corresponds to the objective function in Eq. (15). Therefore, minimizing the objective function of the global model $\epsilon_{\theta_g}$ in Eq. (8) is equivalent to minimizing the difference between this objective function and its lower bound:

$$\min_{\theta_0,...,\theta_N} \frac{1}{N} \sum_{i=0}^{N} D_{KL}(p_{\theta_g}(s_t|\mathcal{C}_i)||p_{\theta_p}(s_t|\mathcal{C}_T)) - D_{KL}(p_{\theta_i}(s_t|\mathcal{C}_i)||p_{\theta_p}(s_t|\mathcal{C}_T)), \forall t \in \{0, ..., T\}$$
$$\tag{17}$$

By expanding this expression, Theorem 1 is thus proved.

**Lemma 1** *For each client $i$ and its unlearning set $\mathcal{D}_i$, the objective function in Eq. (10) has an upper bound, and the upper bound is achieved when $p_{\theta_g}(x_j)$ and $x_j$ are independent for all $x_j \in \mathcal{D}_i$, for example:*

$$D_{KL}(p_{\theta_p}(x_j)||p_{\theta_g}(x_j)) \leq D_{KL}(p_{\theta_p}(x_j)||\mathcal{N}(0, I)), \forall x_j \in \mathcal{D}_i \tag{18}$$

The proof of Lemma 1 can be found in Thomas & Joy (2006) and will not be repeated here. Based on Lemma 1 and the proof of Theorem 1, Theorem 2 can be similarly proved.

## A.2 DATASET DETAILS

In the experimental section, we conducted quantitative and visual experiments on the two constructed datasets, Celebs and Artists. Table 3 presents the specific composition of the unlearn set and retain set for the Celebs dataset, while Table 4 shows the corresponding sets for the Artists dataset. Additionally, we have compiled the links to each image into a file, which is provided in the supplementary material.

## A.3 IMPLEMENTATION DETAILS

In our experiments, the pre-trained DM we mainly used is *Stable-diffusion-v1.5* from the *HuggingFace* model repository, which includes a corresponding CLIP text encoder used in our method to extract the word embeddings of concepts. The *Stable-diffusion-v1.5* are pre-trained on the LAION-5B dataset Schuhmann et al. (2022), covering a wide range of image distributions encountered in daily life. All experiments are conducted with four NVIDIA GeForce RTX 3090 GPUs. Regarding specific hyperparameters, the weight $\beta$ in the loss function is set to $1$. The relevant hyperparameters for the diffusion generation process are set to their default values. The number of inference steps is 50, and the guidance scale of the generation is 3.

| Celebrity | | | | |
|---|---|---|---|---|
| **Unlearn Set** | Kate Upton' | 'Chris Hemsworth' | 'Bette Midler' | 'David Bowie' | 'John Oliver' |

| | Celebrity | | | | |
|---|---|---|---|---|---|
| **Unlearn Set** | Kate Upton' | 'Chris Hemsworth' | 'Bette Midler' | 'David Bowie' | 'John Oliver' |
| | 'Justin Timberlake' | 'Aretha Franklin' | 'Elvis Presley' | 'Andy Samberg' | 'Elon Musk' |
| | 'Judy Garland' | 'Avril Lavigne' | 'Nicole Kidman' | 'George Bush' | 'Matt Damon' |
| | 'Amy Poehler' | 'George Clooney' | 'Paul Mccartney' | 'Clint Eastwood' | 'Ed Sheeran' |
| | 'Idris Elba' | 'Bruce Lee' | 'Dwayne Johnson' | 'Amy Winehouse' | 'Dakota Johnson' |
| | 'Amy Schumer' | 'Megan Fox' | 'Ryan Gosling' | 'Anne Hathaway' | 'Margot Robbie' |
| | 'Theresa May' | 'Anna Kendrick' | 'Ben Stiller' | 'Donald Trump' | 'Anna Faris' |
| | 'Jennifer Lopez' | 'Michael Cera' | 'Bob Marley' | 'Matthew Mcconaughey' | 'Barack Obama' |
| | 'Michael Ealy' | 'Drake' | 'Amy Adams' | 'Jensen Ackles' | 'Barry Manilow' |
| | 'Aziz Ansari' | 'Johnny Depp' | 'Countess Vaughn' | 'Mel Gibson' | 'Tom Hiddleston' |
| **Retain Set** | Robert De Niro' | 'Meryl Streep' | 'Ben Affleck' | 'Mariah Carey' | 'Hillary Clinton' |
| | 'Pierce Brosnan' | 'Joe Biden' | 'John Lennon' | 'Emma Stone' | 'Kristen Stewart' |
| | 'Leonardo Dicaprio' | 'Andrew Garfield' | 'Angelina Jolie' | 'Bruce Willis' | 'Bill Clinton' |
| | 'Ronald Reagan' | 'Tom Hanks' | 'Cameron Diaz' | 'Morgan Freeman' | 'Jackie Chan' |
| | 'Rihanna' | 'Gal Gadot' | 'Courteney Cox' | 'Milla Jovovich' | 'Jennifer Aniston' |
| | 'Hugh Jackman' | 'Anjelica Huston' | 'Keanu Reeves' | 'Gwyneth Paltrow' | 'Justin Bieber' |
| | 'Patrick Stewart' | 'Melania Trump' | 'Aaron Paul' | 'Amber Heard' | 'Amanda Seyfried' |
| | 'Tom Cruise' | 'Arnold Schwarzenegger' | 'Natalie Portman' | 'Benicio Del Toro' | 'Octavia Spencer' |
| | 'Bob Dylan' | 'Kim Jong Un' | 'Jared Leto' | 'Adriana Lima' | 'Chris Evans' |
| | 'Kate Winslet' | '.ipynb_checkpoints' | 'Adam Driver' | 'Jennifer Lawrence' | 'Lana Del Rey' |

Table 3: The composition of the unlearn set and retain set for the Celebs dataset.

| | Artist | | | | |
|---|---|---|---|---|---|
| **Unlearn Set** | Fra Angelico | Hendrick Avercamp | John Constable | Stuart Davis | Jean-Michel Basquiat |
| | Paolo Uccello | Artemisia Gentileschi | John Crome | Edward Hopper | David Alfaro Siqueiros |
| | Piero della Francesca | Frans Hals | Théodore Géricault | Maria Sibylla Merian | Norman Rockwell |
| | Carlo Crivelli | Gian Lorenzo Bernini | John Martin | William Henry Hunt | Will Barnet |
| | Sandro Botticelli | Jacob Jordaens | Richard Parkes Bonington | John James Audubon | Philip Guston |
| | Leonardo da Vinci | Pieter Claesz | Thomas Cole | Marianne North | Gerard Sekoto |
| | Filippino Lippi | Francisco de Zurbaran | James Tissot | Harriet Backer | George Pemba |
| | Sebastiano del Piombo | Jusepe de Ribera | Eugène Girardet | Archibald Thorburn | Romare Bearden |
| | Alessandro Allori | Giovanni Battista Gaulli | James Ensor | George Caleb Bingham | Jean Arp |
| | Jacopo Bassano | Rosalba Carriera | Rudolf Ernst | Paul Gauguin | Jackson Pollock |
| | Sofonisba Anguissola | William Hogarth | Kees van Dongen | Natalia Goncharova | Franz Kline |
| | Lavinia Fontana | Thomas Gainsborough | Henri Matisse | Raoul Dufy | Norman Bluhm |
| | Giuseppe Arcimboldo | Pompeo Batoni | Frederic Edwin Church | David Burliuk | Stanisław Szukalski |
| | Orazio Gentileschi | Bernardo Bellotto | Aaron Siskind | Marc Chagall | Rene Magritte |
| | Robert Campin | Joshua Reynolds | Martin Johnson Heade | Maurice Prendergast | Yves Tanguy |
| | Jean Fouquet | Hubert Robert | George Catlin | Boris Kustodiev | Eileen Agar |
| | Gerard David | Thomas Lawrence | Charles-Francois Daubigny | Lyonel Feininger | Georges Braque |
| | Hieronymus Bosch | Francisco Goya | Henri Fantin-Latour | Frida Kahlo | Paul Delvaux |
| | Lucas Cranach the Elder | Franz Xaver Winterhalter | Jules Breton | Johannes Itten | Arshile Gorky |
| | Matthias Grünewald | Jacques-Louis David | Eastman Johnson | Diego Rivera | Lorser Feitelson |
| | Hans Holbein the Younger | Camille Corot | Ivan Shishkin | Tarsila do Amaral | Remedios Varo |
| | Hans Baldung | Nikolai Ge | Edgar Degas | Constantin Brancusi | Hans Bellmer |
| | Pieter Bruegel the Elder | Albrecht Anker | Claude Monet | Fernand Leger | Joseph Cornell |
| | Pieter Brueghel the Younger | Jules Bastien-Lepage | Vincent van Gogh | Fernando Botero | Marcel Duchamp |
| | M.C. Escher | Luke Fildes | Giovanni Boldini | Candido Portinari | Octavio Ocampo |
| | Louis Comfort Tiffany | Beauford Delaney | Alfred Sisley | Andre Derain | Graham Sutherland |
| | Magnus Enckell | George Stubbs | Mary Cassatt | Adolph Gottlieb | Michael Sowa |
| | Annibale Carracci | Henry Fuseli | Odilon Redon | Mary Fedden | Louise Bourgeois |
| | Adam Elsheimer | J.M.W. Turner | Rockwell Kent | Victor Brauner | Jacek Yerka |
| | Jan Brueghel the Elder | Caspar David Friedrich | George Bellows | Andy Warhol | Henri de Toulouse-Lautrec |
| **Retain Set** | Georges Seurat | Lawren Harris | Sonia Delaunay | John Chamberlain | Wayne Thiebaud |
| | Paul Ranson | Giorgio de Chirico | Odd Nerdrum | Lee Bontecou | Julio Le Parc |
| | Ferdinand Hodler | George Inness | Ernst Ludwig Kirchner | Dan Flavin | Lucio Fontana |
| | Maximilien Luce | Ralph Blakelock | Henry Darger | Donald Judd | Peter Max |
| | Albert Marquet | Granville Redmond | Gustave Buchet | John Hoyland | Richard Hamilton |
| | Cuno Amiet | Gustave Caillebotte | Otto Dix | Juan Gris | Peter Blake |
| | Alexandre Benois | Albert Dubois-Pillet | Aaron Douglas | Karl Knaths | Takato Yamamoto |
| | Jacek Malczewski | Paul Signac | Eugène Grasset | Patrick Caulfield | Barkley L. Hendricks |
| | Piet Mondrian | Carl Larsson | Aubrey Beardsley | Ellsworth Kelly | Jeff Koons |
| | Augustus John | Tsuguharu Foujita | Ivan Bilibin | Patrick Heron | Yves Klein |
| | Theo van Rysselberghe | Eric Ravilious | Beatrix Potter | Raoul De Keyser | Mimmo Rotella |
| | Wassily Kandinsky | Charles Blackman | John Bauer | Robert Indiana | James Lee Byars |
| | Henri-Edmond Cross | Eyvind Earle | Anne Brigman | Howardena Pindell | Felix Gonzalez-Torres |
| | Umberto Boccioni | Alex Colville | Boris Grigoriev | Robert Delaunay | Olafur Eliasson |
| | Maurice de Vlaminck | Peter Doig | Jean Dubuffet | Asger Jorn | James Turrell |
| | Emily Carr | Charles Angrand | George Luks | Ernst Wilhelm Nay | El Anatsui |
| | Koloman Moser | Giacomo Balla | Mary Jane Ansell | Hans Hartung | Takashi Murakami |
| | André Lhote | Marsden Hartley | Thomas Kinkade | Keith Haring | Romero Britto |
| | Thomas Hart Benton | Gino Severini | Richard Lindner | György Kepes | Robert Williams |
| | William Blake | Arthur Dove | Walter Crane | Nan Goldin | Mark Ryden |
| | Arnold Böcklin | Josef Capek | Richard Eurich | Judy Chicago | Simon Stalenhag |
| | Pierre Puvis de Chavannes | Lyubov Popova | Jasper Johns | Howard Finster | Kent Monkman |
| | Gustav Klimt | Willi Baumeister | Sam Gilliam | Yaacov Agam | Njideka Akunyili Crosby |
| | Fernand Khnopff | Josef Albers | Willem de Kooning | Georg Baselitz | Hasui Kawase |
| | Carlos Schwabe | Albert Gleizes | Albert Bierstadt | Adrian Ghenie | Hiroshi Yoshida |
| | Maxfield Parrish | Anni Albers | Sam Francis | Eric Fischl | Kitagawa Utamaro |
| | Roger de La Fresnaye | Ben Nicholson | Richard Diebenkorn | Martin Kippenberger | Katsushika Hokusai |
| | Hilma af Klint | Milton Avery | Helen Frankenthaler | Marlene Dumas | Emily Kame Kngwarreye |
| | Egon Schiele | Yiannis Moralis | Robert Rauschenberg | Anselm Kiefer | Eugène Atget |
| | Charles E. Burchfield | Roy Lichtenstein | Elaine de Kooning | William James Glackens | Karl Blossfeldt |

Table 4: The composition of the unlearn set and retain set for the Artists dataset.

