# OpenReview forum: "Federated Unlearning with Diffusion Models"
_ICLR.cc/2025/Conference — ICLR 2025 Conference Withdrawn Submission_

### Official Review · Reviewer_NYcS · 2024-10-27

**Soundness:** 1
**Presentation:** 1
**Contribution:** 2
**Rating:** 3
**Confidence:** 3

**Summary:**

The paper proposes an unlearning algorithms for diffusion models in federated learning. The unlearning is performed at the local client level where the client proposes to forget about specific samples. It argues that the global model will not relearn the unlearnt knowledge after aggregation at each communication round.

**Strengths:**

The submission proposes an unlearning algorithm in the context of federated learning with diffusion models.

The experiments show some visualizations of the effects of unlearning over image data on the Celebs and Artists datasets.

**Weaknesses:**

It is not clear why minimizing the Frobenius norm of the attention map for the sensitive objective helps with unlearning.

Writing needs to be improved.
Some sentences are difficult to parse.
Some symbols or equations are inaccurate. For instance, in Eq (2), sum from 0 to N means there are N+1 numbers of clients in total. In Theorem 1, the right hand side of Eq. (9) is dependent on i, whereas the left hand side is the sum of losses across all clients.
The format of citations also needs to be fixed throughout the paper.

The definition (or granularity) of unlearning needs to be clarified at the beginning of the paper to set up the context—whether it targets at unlearning at the client level, or local sample level, or some feature level.

I think Theorem 1 and 2 are observations, as opposed to theorems. Lemma 1 is a restatement of previous results without any citations in the main text.

It is not clear if the federated unlearning setup involves any personalization, i.e., if the goal is to learn a single global model that forgets about individual clients’ samples, or to learn personalized models where each client model forgets about that client’s unlearnt data. While in practice, personalization could be achieved simply by performing some local finetuning, the two problems still correspond to different techniques and reasonings.

**Questions:**

I didn’t understand why Theorem 1 indicates that it is not likely for the aggregated global model to recover some unlearnt samples/concepts. Eq. (9) is difficult to parse by itself (dependence on i on the right hand side, H is not defined, etc.). I looked at the proof in the appendix but still don’t understand.

---

### Official Review · Reviewer_GbZ7 · 2024-11-01

**Soundness:** 2
**Presentation:** 3
**Contribution:** 2
**Rating:** 3
**Confidence:** 4

**Summary:**

This paper studies the unlearning problem in federated learning where pre-trained diffusion models are used. Pre-trained diffusion models may generate images that contain sensitive information, so it is very important to erase this unwanted knowledge in FL. The authors propose a method attempting to minimize the difference between the local models (which will be updated to remove information about the forgetting data) and the target model that contains information about the safe domain (minimize contrastive loss between the two attention maps), therefore the local client models and the global model can forget information about the forgetting data.

**Strengths:**

- Motivation is clear and the study problem is important, as pre-trained diffusion models may inadvertently generate images that reveal sensitive information. In the context of FL, it is crucial to effectively remove this unwanted knowledge to ensure privacy and protect against unintended data leakage.
- Experimental results seem effective in forgetting.

**Weaknesses:**

- The theoretical part seems cannot support the claim such as "leverage traditional federated learning methods to address federated unlearning issues".
- It is unclear (the rationale behind) why considering the contrastive loss between the two attention maps, not directly adopting existing machine unlearning methods such as random labeling or assigning to the safe target.
- Current experiments are not enough to support the conclusion.

Please see Questions for details.

**Questions:**

- In the submission, the authors claimed: "However, in the unlearning process, each model has forgotten a portion of its knowledge. During aggregation, it may seem that the knowledge could complement one another, resulting in the global model not actually achieving unlearning. While this reasoning is fundamentally flawed, as unlearning and learning processes are essentially unified." What is the flaw here? It's reasonable that the aggregation can help the knowledge complement each other in FL, for example, evidence in [1].
While the minimization in Theorem 1 supports the claim that unlearning and learning can be viewed as unified processes, it also implies that unlearning requires careful control during aggregation; otherwise, complementary information from each client could counteract unlearning, but the authors suggest that this idea is flawed. Can the authors please explain this?
- The KL divergence is generally not symmetric, from Eq.(17), we cannot get Eq. (13) in Theorem 1.
- Can the authors please explain, why can we not directly apply existing machine unlearning or even federated unlearning methods in this project? To me, the objective, i.e., contrastive loss between the two attention maps, is like the way how existing machine unlearning methods doing, assign another safe label to the forgotten data [2-3].
- Experiments only consider 50% of clients to be forgotten, what if 10% of clients request for unlearning? Would the undesired knowledge complement each other more easily? Also, it would be beneficial to consider different scenarios such as class-level and sample-level forgetting [4].
- Regarding the metric, can the authors provide the FID score as well? To verify the unlearned model didn’t lose the utility. I believe this submission can be benefited from a comprehensive analysis and rigorous reasoning.

[1] Chang, Hongyan, and Reza Shokri. "Bias propagation in federated learning." ICLR 2023.

[2]  Fan, Chongyu, et al. "Salun: Empowering machine unlearning via gradient-based weight saliency in both image classification and generation." ICLR2024.

[3] Heng, Alvin, and Harold Soh. "Selective amnesia: A continual learning approach to forgetting in deep generative models." Advances in Neural Information Processing Systems 36 (2024).

[4] Zhao, Yang, et al. "A survey of federated unlearning: A taxonomy, challenges and future directions." arXiv preprint arXiv:2310.19218 (2023).

---

### Official Review · Reviewer_jHv5 · 2024-11-01

**Soundness:** 2
**Presentation:** 3
**Contribution:** 2
**Rating:** 5
**Confidence:** 3

**Summary:**

This paper studies federated unlearning for diffusion models. The paper proposes a new loss function for federated unlearning and presents some numerical results for the proposed algorithm FedDUL.

**Strengths:**

The paper presents numerical results showing the improvement of their algorithm compared to the previous techniques.

**Weaknesses:**

1- I am not sure from the novelty of the proposed idea compared with the federated unlearning algorithms in the literature. From my point of view, the paper is doing a similar technique of maximizing (locally) the KL divergence and then average the local model.

2- I am not convinced with the loss function in Eq(3), Eq(4), and Eq(8). Maybe the KL divergence of the distiribtuion  $p_{\theta_p}(.|C_i)$ and $p_{\theta_p}(.|C_T)$  are already large. I think it make more sense to maximize the difference [D_KL(on C_i) - \alpha D_KL (C_T)] which means unlearning task C_i and learn the task C_T.

3- The Lemmas and theorems are trivial and don't provide any insights. For example, Theorem 2 can be proven in one line expansion of the KL divergence from Lemma 1. What is the upper bound in Lemma 1? There is always a trivial upper bound on the KL divergence.  What are the implications  and the importance of Theorem 1 and Theorem 2?

**Questions:**

See above

---

### Official Review · Reviewer_ou4w · 2024-11-03

**Soundness:** 1
**Presentation:** 1
**Contribution:** 2
**Rating:** 3
**Confidence:** 3

**Summary:**

In this paper, the authors introduce a federated unlearning problem setting for clients using diffusion models. They propose FedDUL, a federated unlearning method, which consists of two main components: Client Local Unlearning and Server Model Aggregation. In the Client Local Unlearning phase, each client uses local data along with a defined unlearning concept and a corresponding target concept to perform LoRA fine-tuning on the local diffusion model, achieving unlearning of private concepts. The trained parameters are then uploaded to the server. In the Server Model Aggregation phase, the server clusters the received concept word embeddings and performs two rounds of parameter averaging to obtain the global model based on the clustering results, thereby accommodating the unlearning requests of multiple clients simultaneously.

The paper proposes a few theoretical results showing an equivalence between federated learning and federated unlearning. Afterwards, the paper provides quantitative and qualitative experimental results demonstrating the effectiveness of FedDUL. In particular, the experiments show that FedDUL can satisfy the unlearning requests of multiple clients, while preserving the overall performance of the generative model.

**Strengths:**

* The proposed local unlearning technique (Section 3.2) is relatively novel, and well-justified. This technique consists in building a contrastive loss between the attention maps of the target concept and the unlearning concept, and introducing an attention loss  based on the activation map of the unlearning concept.
* The numerical experiments show the advantage of the FedDLU in comparison to other baselines. In particular, the numerical results demonstrate the ability of FedDLU to forget unlearning concepts, while keeping a reasonable performance on the other concepts.

**Weaknesses:**

* The overall writing quality could be improved; there a few typos (e.g., line 053, an novel; line 219, it does not related), the paper does not use parenthesis for in-text citing, some figures are not commented (e.g., Figure 2 and Figure 3).
* The theoretical results and mathematical formulation are far from being sound:
   * Equation 2 is not well defined; what does it mean to solve an optimization for all $t \in \{1, \dots, T\}$? does it mean that we are solving $T$ optimization problem?
   * The optimization problem in (2) aims at maximizing the divergence between two conditional distributions, without having any further constraints. This formulation does not much the goal described in the paper, as it does not constrain the target distribution to have a good generative power on the non-private concepts.
   * Similar problems as above hold also for problem (3).
   * Theorem 1 is trivial.
   * I doubt the correctness of Lemma 1. For example, when $p_{\theta_p}(x_j)$ is the normal distribution, Lemma 1 implies that both $\theta_g$ and $\theta_g$ induce the same distribution. Further justification and explanation is needed.
   * In lines 308--310, the paper claims an equivalence between (2) and (3) without proving it.
   * *nit*: $\sum$ is usually reserved for discrete integration, while $\int$ is used in the non-discrete case.
   * The paper promises **detailed** proofs of the theoretical results, but (in my opinion) does not fulfill this promise.
* I do not really understand the motivation behind the clustered aggregation scheme in Section 3.3. For example, if all clusters have the same size, this aggregation would be equivalent to a normal aggregation.

**Questions:**

* what does it mean to solve an optimization for all $t \in \{1, \dots, T\}$? does it mean that we are solving $T$ optimization problem?
* Could provide a rigorous proof of Lemma 1, in particular how to prove it using the results from (Thomas & Joy; 2006).

---

### Note · Authors · 2024-11-15

I have read and agree with the venue's withdrawal policy on behalf of myself and my co-authors.